# Capsaicin (But Not Other Vanillins) Enhances Estrogen Binding to Its Receptor: Implications for Power Sports and Cancers

**DOI:** 10.3390/life15020208

**Published:** 2025-01-30

**Authors:** Maja Pietrowicz, Robert Root-Bernstein

**Affiliations:** 1Independent Researcher, 37430 Tall Oak Dr., Clinton Township, MI 48036, USA; pietrowiczmaja@gmail.com; 2Department of Physiology, Michigan State University, East Lansing, MI 48824, USA

**Keywords:** capsaicin, acetaminophen, estradiol, estrogen receptor, insulin, glucose transporter, anti-cancer

## Abstract

Capsaicin (CAP), the pain-inducing compound in chili peppers, exerts its effects mainly through the transient receptor potential vanilloid channel 1 (TRPV1), which mediates pain perception and some metabolic functions. CAP has also been demonstrated to improve performance in power sports (but not endurance sports) and does so mainly for females. CAP may also have anti-cancer effects. Many mechanisms have been explored to explain these phenomena, particularly the effects of TRPV1 activation for calcium influx, glucose transporter (GLUT) upregulation and inhibition of insulin (INS) production, but two important ones seem to have been missed. We demonstrate here that CAP binds to both INS and to the estrogen receptor (ESR1), enhancing estradiol binding. Other TRPV1 agonists, such as vanillin, vanillic acid and acetaminophen, have either no effect or inhibit estrogen binding. Notably, TRPV1, ESR1 and INS share significant regions of homology that may aid in identifying the CAP-binding site on the ESR1. Because activation of the estrogen receptor upregulates GLUT expression and thereby glucose transport, we propose that the observed enhancement of performance in power sports, particularly among women, may result, in part, from CAP enhancement of ESR1 function and prevent INS degradation. Chronic exposure to CAP, however, may result in downregulation and internalization of ESR1, as well as TRPV1 stimulation of glucagon-like peptide 1 (GLP-1) expression, both of which downregulate GLUT expression, thereby starving cancer cells of glucose. The binding of capsaicin to the ESR1 may also enhance ESR1 antagonists such as tamoxifen, benefiting some cancer patients.

## 1. Introduction

Capsaicin (CAP), trans-8-methyl-N-vanillyl-6-nonenamide, is a vanilloid compound in chili peppers that is responsible for their hot flavor. It is structurally similar to vanillin (VAN) and vanillic acid (VAN ACID), the main flavor components of vanilla, as well as the analgesic acetaminophen (ACET) (Figure 1).

CAP is known to modulate glucose metabolism through both transient receptor potential vanilloid channel 1 (TRPV1)-dependent and TRPV1-independent mechanisms. In most, but not all, in vitro cellular systems, CAP stimulates glucose uptake via upregulation of glucose transporters (GLUTs), especially GLUT1 and GLUT4, through 5′-adenosine monophosphate protein kinase (AMPK) activation, which in turn stimulates production of Activator Protein 1 (AP-1), a collective term referring to dimeric transcription factors composed of Jun, Fos or activating transcription factor (ATF). CAP also increases expression of TRPV1, which increases influx of calcium into cells and acts as an integrator for heat and pain (nociception) sensation and regulation of body temperature; it also decreases oxygen consumption by mitochondria by decreasing oxidative phosphorylation pathways [1]. Animal studies demonstrated that CAP decreased insulin (INS) release via increased release of glucagon-like peptide 1 (GLP-1), an INS- and glucagon-release inhibitor. CAP also increased INS sensitivity, probably by means of increased expression of GLUT2, GLUT4 and TRPV1. Unlike the in vitro cellular studies, mitochondrial oxidative phosphorylation and overall mitochondrial function were found to have increased [1]. Figure 2 summarizes the main effects of CAP.

In human studies, these CAP effects have been found to increase INS sensitivity, decrease INS resistance, improve blood glucose levels and increase resting energy expenditures [1]. Controlled human trials involving patients with metabolic syndrome have also established that CAP lowers total cholesterol, particularly low-density lipoproteins, by stimulating the preferential metabolism of lipids via an AMPK-mediated mechanism [2], and additionally, in women only, it lowers triglycerides [3]. TRPV1 also mediates lipid metabolism. These studies suggest that CAP may be an effective, natural means of helping type 2 diabetics and those with metabolic syndrome to regulate hyperlipidemia and blood glucose levels and to gain better control over the progression of their disease [4].

CAP also influences some types of physical performance and endurance, although, oddly, these effects have not been studied in metabolic syndrome or diabetic patients who might particularly be expected to benefit. In well-controlled trials involving healthy volunteers, ingestion of CAP or capsiate (a CAP agonist—see Figure 1) in doses of about 12 mg ingested 45 min to an hour prior to exercise had no significant effect on aerobic endurance but significantly improved anaerobic performance involving short bursts of activity such as sprinting and weight-lifting that involved muscular power endurance. CAP also decreased the perception of fatigue following muscular endurance tests but not aerobic endurance tests such as long-distance running [5].

Notably, at least some effects of CAP vary by sex and may be mediated by estrogens [6]. The CAP-associated decrease in triglycerides experienced only by women, mentioned above, is one example. Pain sensation is another [7]. Another is that CAP stimulates the synthesis and expression of TRPV1 in an estrogen-dependent manner [8], thereby enhancing its metabolic effects. These sex hormone effects also affect TRPV1-mediated pain sensation; while testosterone inhibits TRPV1 expression, estrogens increase TRPV1 expression so that a four-fold increase in CAP concentrations is required to induce equivalent pain sensations in male rats compared with female rats [9]. Additionally, pain sensitivity in women varies as a function of their menstrual cycle such that higher estrogen levels are associated with greater pain sensitivity [10,11]. One would therefore expect that the menstrual cycle might affect female sports performance, which appears to be true for sports involving the same types of anaerobic or power generation associated with CAP enhancement described above: “The current evidence suggests a variable association between menstrual cycle and a few performance-related outcomes, such as endurance or power resistance,” but not aerobic endurance sports [12]. Notably, the menstrual cycle significantly modified competitiveness, feelings of psychological well-being and decision-making skills [12], again factors associated with the decreased perception of fatigue noted above as a result of CAP ingestion.

The present research was stimulated by recognition that CAP affects glucose metabolism via its effects on INS and GLUT and that estrogens also affect glucose metabolism via effects on INS and GLUT.

Given the overlap in CAP and estrogen effects on glucose metabolism, we decided to explore whether CAP interacted directly with estrogens, INS and their receptors. Estrogen receptors (ESR1s) modulate INS-stimulated glucose uptake in adipose and muscle tissue [13,14], in part by stimulating GLUT2 and GLUT4 synthesis and membrane expression, resulting in increased cellular uptake of glucose [15]. Similarly, activation of the INS receptor by INS increases GLUT (especially GLUT 4) insertion into the cell membrane, again increasing glucose transport into the cell. Increased cellular glucose then provides negative feedback on INS synthesis and release, helping to regulate the system [16]. If CAP were to stimulate estrogen and/or INS activity, then the observed effects of CAP on increased glucose transport and increased non-aerobic exercise performance might be explained, as well as the enhanced effects mentioned above for obese individuals. If CAP affects estrogen activity, it might further explain some of the sex bias in CAP’s observed effects. For example, estrogens bind directly to INS, inhibiting binding to the INS receptor (IR), which may contribute to the observed variations in INS sensitivity observed during the menstrual cycle and also upon use of estrogen-containing birth control, as well as in gestational diabetes [17]. Crucial to this hypothesis is the observation that CAP can transit the cell membrane in order to interact not only with its binding site on the TRPV1 receptor but also with ERs that are localized intracellularly [18]. Some of the key, established interactions between estrogens, INS and glucose are summarized in Figure 3.

In summary, the purpose of this study is to investigate some novel sex-hormone-related mechanisms of capsaicin and other vanilloid compounds that may influence their acute effects on glucose metabolism. In particular, we hypothesize that the observed sex-dependent differences in CAP effects may be mediated by the fact that capsaicin’s sex-related metabolic effects may be due to direct interactions with insulin, estrogens and/or their receptors.

## 2. Materials and Methods

Capsaicin, vanillin, vanillic acid, acetaminophen, estradiol, estriol, 5-pregnan-3β-OL-2O-one, progesterone, 4-androstene-3,17-dione and insulin (human recombinant) were all obtained from Sigma-Aldrich (St. Louis, MO, USA). Estradiol (E2)-6-HRP (NOV-ND-R0951) was acquired from Enzo Biochem (Farmingdale, NY, USA). Estrogen receptor alpha (ER alpha) recombinant protein was obtained from Invitrogen-Thermofisher (Waltham, MA, USA).

Enzyme-linked immunosorbent assay (ELISA) was used to investigate whether CAP acts as a competitive inhibitor or allosteric regulator in the presence of estrogen binding to the ESR1. An estrogen linked to horseradish peroxidase (EST-HRP) served as both the source of estrogen and a means by which to evaluate changes in the binding affinity of estrogen to its receptor in the presence of other ligands, such as CAP and ACET. To begin, the ESR1 was dissolved in a 1:1 solution of water to ethanol at a 1/100 dilution. Then, 100 μL of this solution was pipetted in duplicate into 10 wells of a Costar round-bottomed 96-well ELISA plate and incubated for one hour. Two wells in each row received only the ethanol–water solution and acted as controls. Any excess receptor was triply washed out using a 1% Tween 20 solution (in phosphate buffer) and a plate washer. Then, 200 μL of blocking agent (2% polyvinyl alcohol in phosphate buffer) was added to every well, incubated for an hour and then triply washed. Next, CAP was dissolved at a concentration of 0.1 mg/mL ($3.27\times 10^{-4}M$) in a 1:1 solution of water to ethanol which acted as the base solution for EST-HRP dilution. The preparation of the EST-HRP dilutions was also carried out in a 0.01 mg/mL ($3.27\times 10^{-5}\;M$) and 0.001 mg/mL ($3.27\times 10^{-6}\;M$) CAP base solution to investigate the magnitude by which varying levels of CAP can shift the binding curve. A base solution of 1:1 water to ethanol without CAP served as the control. Beginning at a 1/100 dilution, the EST-HRP was then diluted by thirds ten times. An amount of 100 μL of each dilution was added to a well, incubated for an hour and triply washed. Finally, 100 μL of ABTS reagent was added to each well, incubated for an hour and the plate was read at 405 nm in a Spectramax UV-VIS scanning spectrophotometer.

The same competitive-binding ELISA procedure was used to investigate the effect of other ligands on the binding affinity of estrogen to the ESR1. The other ligands investigated were vanilloids with a similar structure to CAP, specifically VAN and VAN ACID, and known competitive inhibitors of the receptor, specifically ACET. These compounds were also tested against the same hormones as CAP using UV spectroscopy to investigate their binding activity.

UV spectroscopy in a crystal plate was used to investigate whether CAP has specificity for estrogen alone or whether it can bind to other naturally occurring hormones. To start, CAP was dissolved in a 1:1 solution of water to ethanol at 1 mg/mL ($3.27\times 10^{-3}\;M$) and then diluted by thirds eight times. Next, the hormones were dissolved in a 1:1 solution of water to ethanol at a 1/100 dilution. An amount of 50 μL of each CAP dilution was added to 3 wells of a crystal plate, with one well serving as the control, with another 50 μL of 1:1 water to ethanol added, and the other two serving as experimental duplicates, with 50 μL of the prepared hormone solution added. Then, two additional wells were prepared to serve as controls: one containing 100 μL of 1:1 water to ethanol and the other containing 50 μL of prepared hormone solution and 50 μL of 1:1 water to ethanol. Finally, the plate was incubated for 1 h and read in a Spectramax UV-VIS scanning spectrophotometer, spanning wavelengths from 190 to 400 nm. Data analysis involved the subtraction of observed absorbance values from expected absorbance values generated from the control wells ([CAP + 1:1 solution] + [hormone + 1:1 solution] − [1:1 solution]) and plotting them as a function of CAP concentration. Because the solvent (1:1 solution of water–ethanol) was used as a control and subtracted from all wells, the resulting data reflected only the absorbances of the solute compounds. Since each compound has a different lambda maximal absorbance, analysis of the resulting data required finding one common wavelength at which the additions and subtractions of the spectra could be carried out and at which the absorbance of each compound was maximized. In practice, the best common wavelengths were found to be between 205 and 210 nm. These wavelengths also suggest that the dominant form of binding involves hydrogen and/or ionic bonding rather than interactions such as charge transfer complexing that would be dominant at higher wavelengths around 270–280 nm.

Data were analyzed and graphed using Microsoft Excel and diagrams were created on Microsoft PowerPoint (Redmond, WA, USA).

Similarity searches involving TRPV1, insulin and the estrogen receptor (SwissProt accession numbers provided in the relevant figures) were carried out using LALIGN (https://www.ebi.ac.uk/jdispatcher/psa/lalign accessed 14 February 2024) with the following parameter settings optimized for finding significant alignments in short sequences of proteins: BLOSUM80, Gap open −12.0, Gap extended −2.0, E value 1.0.

## 3. Results

Because previous research demonstrated that estrogens bind to INS at concentrations relevant to estrogen-overproduction syndromes [17] and because CAP affects glucose metabolism in a sex-dependent manner (see Introduction), our first experiment was to determine whether vanilloid compounds bound to INS using UV spectroscopy, a technique for determining non-covalent chemical interactions that has been well validated by other physicochemical methods such as nuclear magnetic resonance spectroscopy, circular dichroism studies, etc. [17,19,20,21,22,23]. The results are shown in Figure 4. CAP bound to INS with a binding constant of about 4 µM, while the other vanilloid compounds showed non-significant binding, with the possible exception of VAN ACID. Notably, a typical experimental dose of CAP is 12 mg (see Introduction) which, diluted into 6 L of blood, would yield a concentration of 4 µM so that at that dosage, about half of the INS molecules would be bound to CAP. CAP did not bind significantly to the IR except at concentrations well beyond those physiologically possible (Figure 5). Because CAP binds to INS, it was possible that it might antagonize INS binding to its receptor. CAP does not, apparently, alter INS binding to its receptor. Two types of ultraviolet spectroscopy experiments were performed. In one, INS (1 µM) and the IR were kept constant and the concentration of CAP increased from 0.1 µM to 100 µM. In the other, the CAP (25 µM) and IR were kept constant and the concentration of INS increased from 10 nM to 10 µM. In neither experiment was any significant change in the INS binding curve observed.

Since estradiol and CAP both bind to INS, suggesting a common binding motif, we tested whether CAP might also bind to the ESR1. UV spectroscopy was used once again and the results are shown in Figure 5. CAP did bind to the ESR1 with what appears to be a high affinity (Kd = 0.9 µM) and a lower affinity (Kd = 10 µM) site. Not surprisingly, estradiol did not bind to the IR. Other vanilloid compounds also bound to the ESR1 but with lower affinity than CAP (Figure 6). None bound to the IR.

The next question that needed to be answered was whether CAP competed with estradiol for binding to ESR1. For this study, an enzyme-linked adsorption assay was employed that used EST-HRP. The results are shown in Figure 7. CAP did not compete with estradiol for binding to ESR1. On the contrary, CAP enhanced binding of estradiol to its receptor in a concentration-dependent manner that peaked at a half-log leftward shift in the binding curve, with enhancement even at the lowest concentrations of estradiol and CAP.

The same EST-HRP assay using other vanilloid compounds demonstrated that these additional compounds did not enhance estradiol binding to its receptor. VAN had no significant effect on estradiol binding (Figure 8), while both VAN ACID and ACET antagonized estradiol binding (Figure 9 and Figure 10). Once again, the antagonistic effect was evident even at very low concentrations of estradiol.

Experiments were also performed to test whether the enhancement or antagonism observed in estradiol binding to the ESR1 might be due to vanilloid compounds binding directly to estradiol or to related steroidal hormones, perhaps protecting them from oxidation or forming a complex that prevented the estradiol binding to its receptor. UV spectroscopy was employed once again and the results are summarized in Table 1. No vanilloid compound bound to a measurable degree to estradiol or any other steroidal hormone tested.

To summarize, capsaicin, like estradiol, bound to insulin but not the insulin receptor. However, the binding of capsaicin to insulin had no observable effect on insulin’s binding to its receptor at reasonable physiological concentrations of the compounds. Capsaicin also bound to the estrogen receptor, enhancing estradiol binding to it by approximately a half of a log unit. Neither vanillin nor acetaminophen bound significantly to insulin and weakly to the estrogen receptor whereas vanillic acid bound almost as well as capsaicin to both insulin and the estrogen receptor. Vanillic acid, however, inhibited, rather than enhanced, the binding of estradiol to its receptor, as did acetaminophen (despite its apparently weak binding to the receptor). Vanillin had no effect on estradiol binding to the estrogen receptor. Additionally, capsaicin, vanillin, vanillic acid and acetaminophen did not bind directly to estradiol.

These results are compatible with an allosteric mechanism for capsaicin enhancement of estradiol binding to the estrogen receptor and for ACET and VAN ACID inhibition of binding. The plausibility of the existence of an allosteric site was tested using two types of similarity searches. First, it is well established (see Introduction) that vanilloid compounds such as CAP bind to TRPV1. Above, we demonstrated that CAP and similar compounds bind to ESR1. TRPV1 and the ESR1 share a very large region of homology, as well as a number of smaller regions (Figure 11). Thus, the potential for both proteins to bind common ligands is supported by shared sequences. Additionally, in light of the evidence that CAP binds to both the ESR1 and to INS, a sequence comparison of INS and the ESR1 also reveals significant similarities (Figure 11), which again suggest the possibility of shared ligand binding. Most importantly, TRPV1, ESR1 and INS all share an additional set of common homologies (Figure 12), which may represent the most likely binding sites for vanilloid compounds.

## 4. Discussion

The results reported here suggest that CAP may cause its female-skewed metabolic effects (see Introduction) via a combination of actions on INS activity, estrogen activity and their combined regulation of glucose metabolism. In particular, we demonstrate that CAP, VAN ACID and ACET bind directly to the estrogen receptor but that the binding itself can have different effects. CAP binding enhances estradiol binding to the estrogen receptor while VAN ACID and ACET binding inhibit estradiol binding. Vanillin did not bind significantly to the estrogen receptor nor did it affect estradiol binding to the receptor. None of these compound bound to estradiol itself nor to the INS receptor but all bound to INS. Notably, however, the binding of CAP and the other compounds to INS did not significantly affect INS binding to its receptor. These results suggest that the ability of CAP to modulate INS activity is therefore mediated through the estrogen receptor rather than either INS itself or the INS receptor.

The binding of CAP and related vanilloid compounds to the estrogen receptor and the subsequent alteration in estradiol binding to the receptor suggest that CAP may act as an allosteric modulator of estradiol activity. CAP is known to bind to TRPV1 and we have demonstrated binding to INS and the estrogen receptor, as well as provided a possible way of identifying the location of this putative allosteric region on the estrogen receptor by looking for similarities between the sequences of the three proteins. Figure 11 and Figure 12 show that there are several regions displaying a high degree of similarity that are shared by these proteins. These shared regions, whether pair-wise or involving all three proteins, may represent evolutionarily conserved modules that have been selected for their ability to network diverse ligand interactions into functional interactomes across receptor and hormone classes [24,25].

However, we must be circumspect in interpreting our results since the mechanism we have hypothesized here is only one of many that interact as a result of CAP administration. Considering the ways in which estrogens, vanilloids and INS interact to regulate energy metabolism, especially through control of glucose utilization (see Introduction), reveals a more complex set of effects that are likely to result from CAP binding to INS and the ESR1 (Figure 13). This network of interactions is consistent with previous research studies summarized in Table 2 that have characterized the effects of CAP on glucose metabolism, fat metabolism and exercise by upregulating serum INS levels and sensitivity, thereby decreasing serum glucose levels [26,27]. However, previous research was not able to provide a clear mechanism to explain these effects, and previous investigators have attributed the metabolic affects to both TRPV1-independent and TRPV1-dependent mechanisms. Proponents for TRPV1-dependent mechanisms suggest that TRPV-1 activation stimulates GLP-1 release, increasing INS sensitivity and stimulating INS secretion from pancreatic β-cells [27]. On the other hand, TRPV1-indepndent mechanisms call on capsaicin’s ability to induce the phosphorylation of substrates such as targets in the AMPK pathway and ROS. These findings further support our assumption that the metabolic effects of capsaicin occur as a result of a network of interactions to which we have now added yet another set involving estrogen receptors.

Referring back to the Introduction and Figure 3, binding of estrogens to the ESR1 results in localization to the cell nucleus and activation of gene transcription for INS synthesis. The enhancement of estradiol binding to its receptor can be predicted to enhance INS synthesis as well as glucose transport into cells. CAP binding to TRPV1 also increases calcium flux into cells, increasing metabolism, muscle contractility and AMPK activity. AMPK stimulates AP-1 activity, increasing gene transcription for TRPV1, GLUT (especially GLUT2 and GLUT4) and the IR, thereby, again, increasing glucose transport into cells. These multiple pathways for increasing glucose flux might explain both the ability of CAP to lower blood glucose levels as well as how it enhances short-term power production but not long-term endurance. On the other hand, CAP indirectly represses INS synthesis (probably due to downregulation of the systems just mentioned after chronic or high-dose acute stimulation), and also stimulates GLP1 synthesis and release, which directly regulate INS synthesis. CAP binding to INS may protect INS from degradation, increasing available concentrations in the short term. However, increasing concentrations of intracellular glucose cause a decrease in INS release, lowering IR activation and thus GLUT expression, thereby creating a negative feedback system. The effects of downstream negative feedback systems on GLUT expression might be particularly important in understanding why CAP does not benefit endurance.

### 4.1. Testable Predictions

The schema laid out in Table 2 and Figure 13 and explained in the previous paragraphs leads to a series of testable predictions that may help to validate or invalidate the novel integration of mechanisms of vanilloid activity proposed here.

We observed that both VAN ACID and ACET (paracetamol) impaired estradiol binding to the ESR1. This mechanism is consistent with the many studies that have linked ACET exposure to impaired steroidogenesis and reproductive development in both male and female animals and human cells (e.g., [33,34,35,36]). These effects undoubtedly involve multiple mechanisms and it is not our intent to attempt to summarize these here, but only to suggest that the direct effect of ACET on estrogen activity (and therefore possibly testosterone as well) may be involved. Our results suggest that VAN ACID may have similar effects to those of ACET on the reproductive system but studies involving reproductive hormones and VAN ACID seem to have focused thus far on its effects on improved osteoblast function [37,38] rather than reproductive development and function. It appears that ACET, like VAN ACID, may also alter bone growth. However, reported effects are complex with impairment of load-induced bone formation but prevention of menopause-associated bone loss [39,40]. These results would be consistent with the gender-associated mechanism we have proposed here, and further studies might be able to test whether these reproductive and bone growth effects are associated with alterations in ESR1 function by these compounds.

Another set of testable predictions concerns new explanations of some of the observed effects of CAP on cancers. While many studies (mostly in vitro) suggest that CAP can be an effective anti-cancer agent, other studies suggest that it is a “two-edged sword” that, at some dosages or in some cancers, can promote cell growth and metastasis [41,42]. Proposed mechanisms of the cancer-related effects of CAP are listed in Table 3. Many papers have discussed its ability to stimulate apoptosis by increasing the expression of Bax, by upregulating the cleavage of procaspase- 3 to caspase-3 or by upregulating the cleavage of poly (ADP-ribose) polymerase (PARP) [43,44,45]. Others provide evidence of CAP’s ability to inhibit vascular endothelial growth factor (VEGF) expression, promote cell cycle arrest and suppress ROS production to inhibit tumor angiogenesis, cell proliferation and tumor metastasis, respectively [46,47,48]. Moreover, some authors even suggest that CAP has a selectivity for cancer cells which results in their sensitization, making them more receptive to chemotherapeutic agents [43]. Most interestingly, there were two studies that provide evidence of CAP’s ability to modulate hormone expression which outright prevents the formation of or ameliorates histological changes in tumors. One listed evidence for an increase in estradiol and progesterone expression that blocked the formation of mammary tumors; the other provided evidence for the downregulation of androgen expression which improved histological changes in the subject’s prostate [49,50]. 

Our observations on CAP enhancement of estradiol binding to the ESR1 predict two previously unsuspected mechanisms by which CAP may influence cancer cell growth. One is the enhancement of estrogen binding to the ESR1 which, acutely, may stimulate estrogen-sensitive cancers but chronically and at higher CAP dosages would likely downregulate ESR1 expression, thus producing a tamoxifen-like effect. In this context, it would be extremely interesting to determine whether CAP might enhance tamoxifen (or other ESR1 antagonists) binding to the ESR1. A second mechanism may also be at work. CAP acutely increases GLUT expression allowing increased glucose transport into cells, but chronic dosing activates several negative feedback systems (Table 2 and Figure 13) that decrease GLUT expression and therefore glucose transport. Cancer cells are known to require unusually large amounts of glucose compared with non-cancerous cells [45], suggesting that high-dose, chronic CAP might be effective in downregulating GLUT expression, thereby inhibiting cancer cell metabolism.

### 4.2. Limitations of This Study

The preceding set of predictions reveal some of the many limitations of this study, which include, first of all, the evident fact that the systems that are affected by VAN-like compounds are complex, intertwined and involved with many physiological functions. Thus, the mechanism that we have proposed must, perforce, be only one of many in operation in any one of these systems. To what degree our mechanism is primary, and, if so, under what physiological conditions it occurs, will require further research. A second limitation is the fact that we have relied upon an in vitro assay of estradiol binding which, while it has the advantage of being isolated from other interactions and has yielded results consistent with in vivo experiments conducted by others, may not reflect CAP effects in the more complex systems present in living cells. This, too, will require further experimentation. Perhaps the most important limitation is that, in order to interpret our results and its mechanistic implications, we have relied on previous research demonstrating the physiological and cellular effects of CAP on metabolic functions and physical activity. We believe that our results are consistent with these previous results but that they also explain some of the effects that CAP and other vanillins have on metabolism that have previously puzzled researchers.

## 5. Conclusions

Previous research has demonstrated that capsaicin modulates glucose metabolism through both transient receptor potential vanilloid channel 1 (TRPV1)-dependent and TRPV1-independent mechanisms. In particular, CAP binding to TRPV-1 results in upregulation of glucose transporters, which in turn increase glucose uptake. CAP also increases INS sensitivity and release of glucagon-like peptide 1, which further modulate energy metabolism. A largely independent set of research studies demonstrate that CAP affects sports performance and pain perception, particularly in women. And a third set of research studies has demonstrated that estrogens can bind directly to INS, modulating INS activity. Our research integrated these three sets of studies to hypothesize that the sex-related differences in CAP modulation of insulin activity may be due to direct interaction of CAP with INS, estrogens or their receptors. Our results suggest that while CAP binds directly to INS, this binding does not have any effect on INS binding to its receptor nor does CAP bind to the INS receptor itself. Therefore, CAP probably does not affect INS activity in any direct manner. CAP also does not bind to estrogens. However, CAP does bind directly to the estrogen receptor and this binding enhances estradiol binding to the estrogen receptor. The estrogen receptor, in turn, is well known to stimulate INS gene expression, thereby providing a mechanism for CAP to affect INS production and activity in a sex-dependent manner.

Some other vanilloid compounds also bound with somewhat less affinity than CAP to the estrogen receptor and INS but not to the INS receptor or to estradiol. Vanillin did not bind significantly to INS, the INS receptor, estradiol or the estrogen receptor. These were vanillic acid and acetaminophen. The differences in binding affinities to the ESR1 by the various compounds did not directly predict whether they would enhance, inhibit or have no effect on estradiol binding. In fact, both VAN ACID and ACET inhibited estradiol binding to the estrogen receptor at the same concentrations as CAP enhanced this binding. These results suggested that CAP and the other vanilloid compounds might produce non-TRPV1-mediated effects by means of an allosteric receptor mechanism. The plausibility of such a mechanism was shown by demonstrating the existence of significant similarities between TRPV1, the estrogen receptor and INS. The existence of these shared similarities along with our demonstration of binding of CAP, VAN ACID and ACET to the estrogen receptor provide evidence for an allosteric binding site on the estrogen receptor that can modulate estradiol binding. This modulation would then, in turn, modulate activation of INS gene transcription, INS concentrations and, in turn, glucose entry into cells.

CAP modulation of INS and glucose activity in a sex-dependent manner has implications not only for understanding the ability of CAP to improve female, but not male, performance in power sports but also for the use of CAP as a adjuvant for some types of cancer therapies. CAP has been found previously to be an effective anti-cancer adjuvant for some cancers at some dosages but not for other types of cancers or for all patients. It is not known how CAP functions as an anti-cancer agent but we hypothesize that it may be more effective for people of one sex than the other (which has not, apparently, been investigated) and for estrogen-sensitive cancers (perhaps by enhancing tamoxifen and other estrogen blockers), and may have direct effects on cancer cell survival by altering glucose uptake. Thus, testing our hypothesis has provided evidence of a novel mechanism by which CAP modulates INS activity by means of allosteric enhancement of estrogen binding to its receptor, which has implications for understanding observed CAP effects in both metabolism and cancers.

## Figures and Tables

**Figure 1 life-15-00208-f001:**
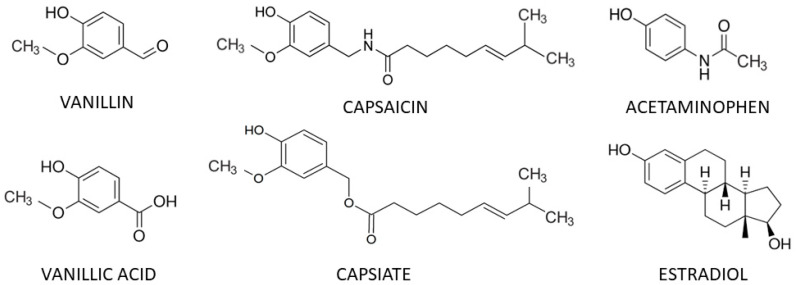
Structures of some vanilloid compounds, acetaminophen and estradiol.

**Figure 2 life-15-00208-f002:**
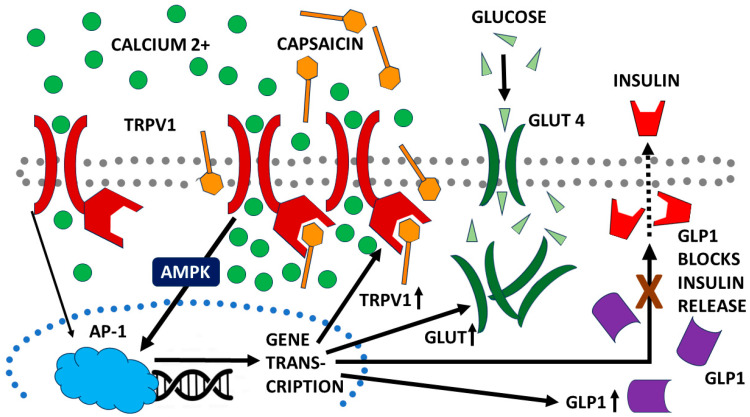
Schematic diagram of some of the most important, known mechanisms by which capsaicin (CAP) modulates energy metabolism. Transient receptor potential vanilloid channel 1 (TRPV1); glucose transporter (GLUT); glucagon-like peptide 1 (GLP1); 5′-adenosine monophosphate protein kinase (AMPK); Activator Protein 1 (AP-1), a collective term referring to dimeric transcription factors composed of Jun, Fos or activating transcription factor. Arrows show the direction of the activity stimulated by each response.

**Figure 3 life-15-00208-f003:**
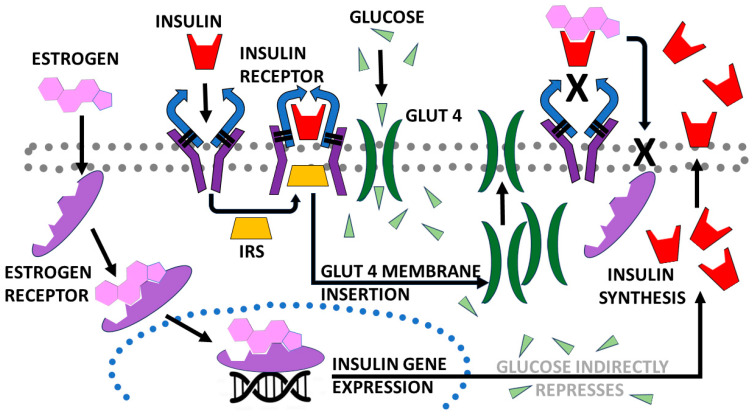
Schematic diagram of some of the key, established interactions between estrogens, insulin (INS) and glucose. Glucose transporter type 4 (GLUT 4); insulin response substrate (IRS). X indicates that the complex of INS with estrogen blocks IR activity. Arrows show direction of activity provoked by each response.

**Figure 4 life-15-00208-f004:**
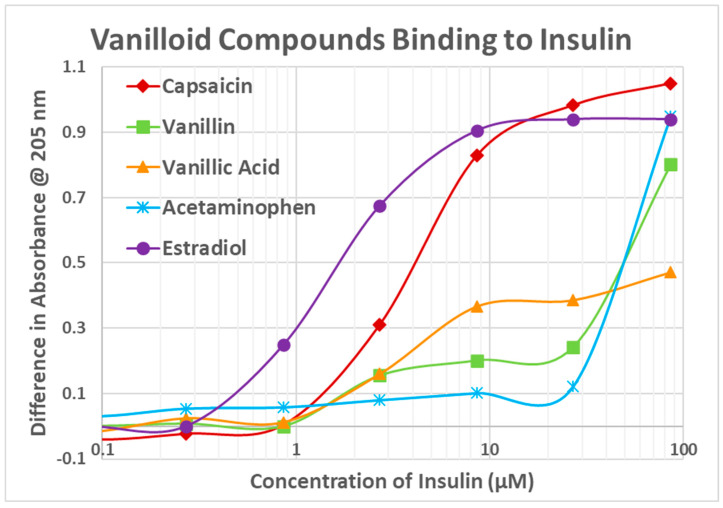
Vanilloid compound binding to insulin as determined by ultraviolet spectroscopy.

**Figure 5 life-15-00208-f005:**
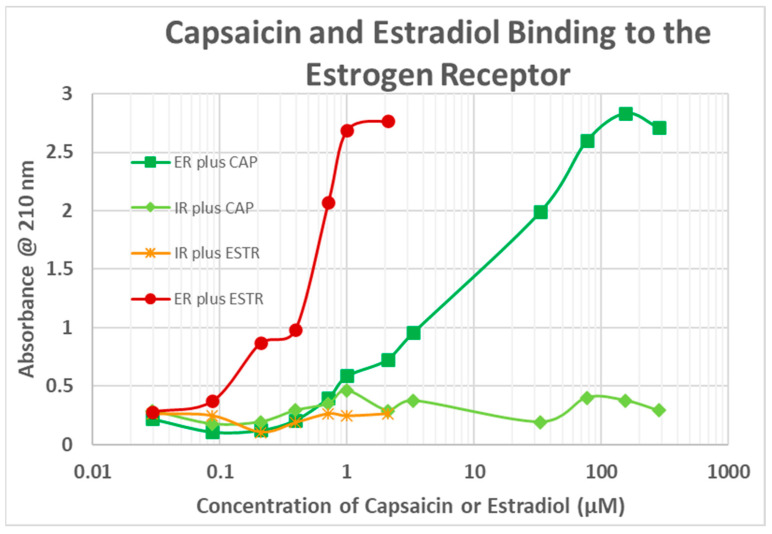
Capsaicin (CAP) and estradiol (ESTR) binding to the estrogen receptor (ER) and the insulin receptor (IR) as determined by ultraviolet spectroscopy.

**Figure 6 life-15-00208-f006:**
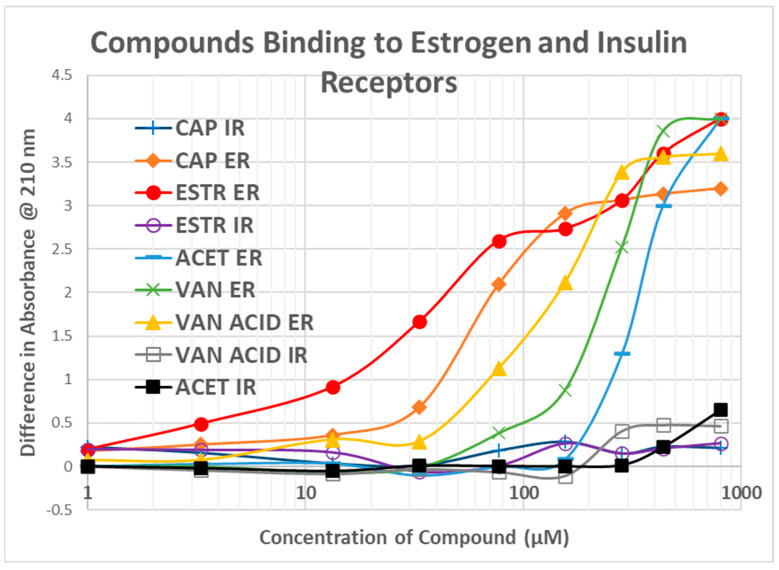
Capsaicin (CAP), vanillin (VAN), vanillic (VAN ACID), acetaminophen (ACET) and estradiol (ESTR) binding to the estrogen receptor (ESR1) and insulin receptor (IR) as determined by ultraviolet spectroscopy.

**Figure 7 life-15-00208-f007:**
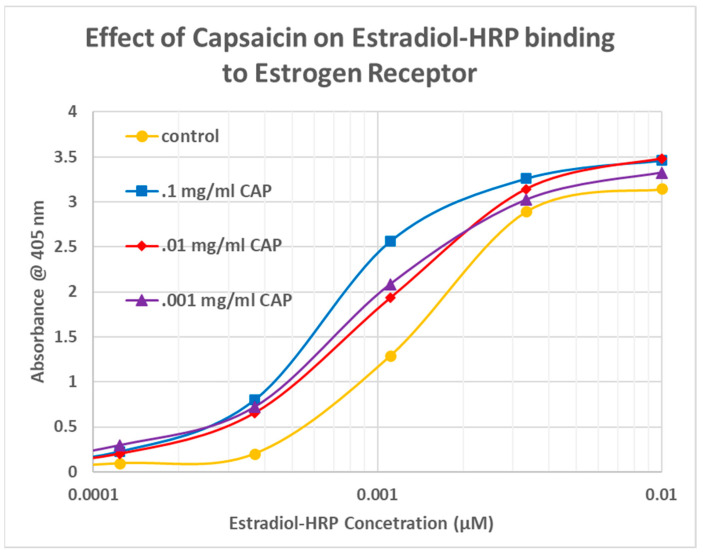
Effect of adding capsaicin (CAP) to EST-HRP (estradiol conjugated to horseradish peroxidase) on the binding of EST-HRP to the estrogen receptor. CAP increases binding of estradiol to its receptor. The control is EST-HRP binding to the estrogen receptor in the absence of any CAP.

**Figure 8 life-15-00208-f008:**
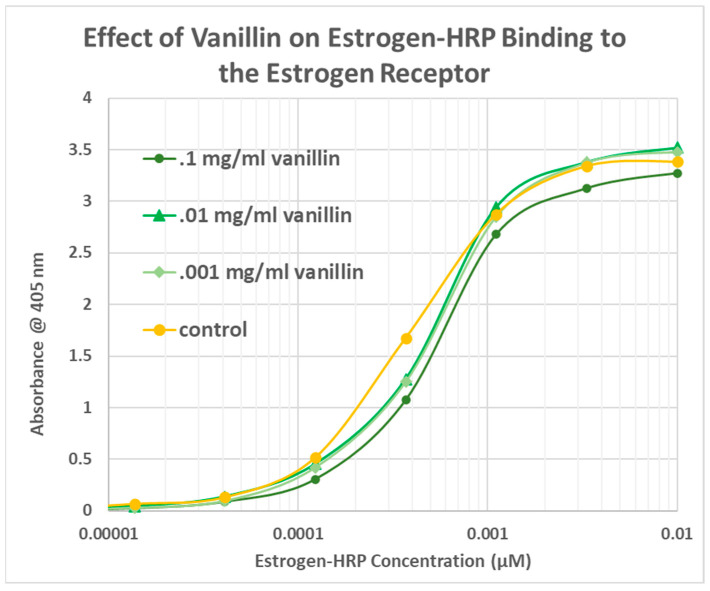
Effect of vanillin on the binding of estrogen conjugated to horseradish peroxidase (EST-HRP) to the estrogen receptor. Vanillin slightly decreased estradiol binding. The control is EST-HRP binding to the estrogen receptor in the absence of any vanillin.

**Figure 9 life-15-00208-f009:**
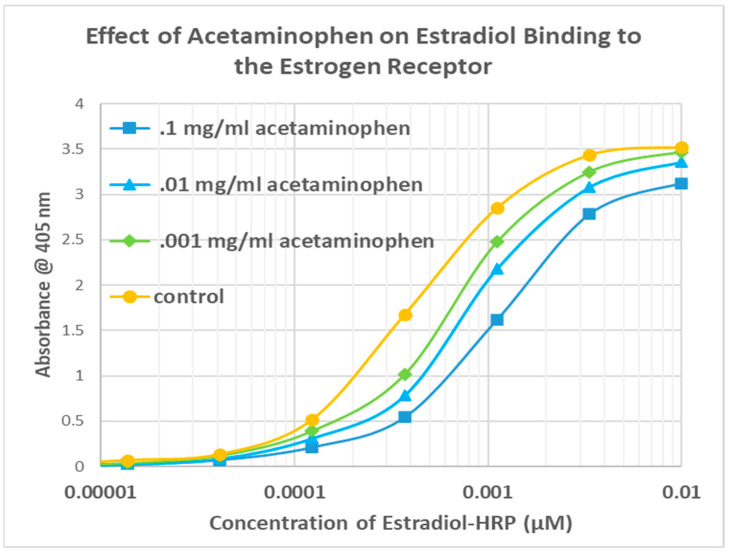
Effect of acetaminophen on estradiol conjugated to horseradish peroxidase (EST-HRP) on binding to the estrogen receptor. Acetaminophen significantly decreased (by up to half a log unit) estradiol binding to its receptor. The control is the binding or EST-HRP to the estrogen receptor in the absence of any acetaminophen.

**Figure 10 life-15-00208-f010:**
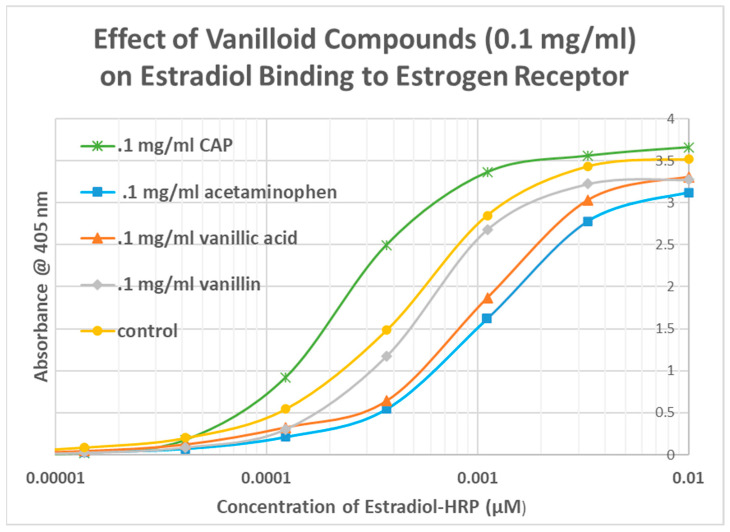
Comparison of the effects of different vanillin-like compounds on the binding of estradiol conjugated to horseradish peroxidase (EST-HRP) to the estrogen receptor. The control is the binding of EST-HRP to the estrogen receptor in the absence of any other compound.

**Figure 11 life-15-00208-f011:**
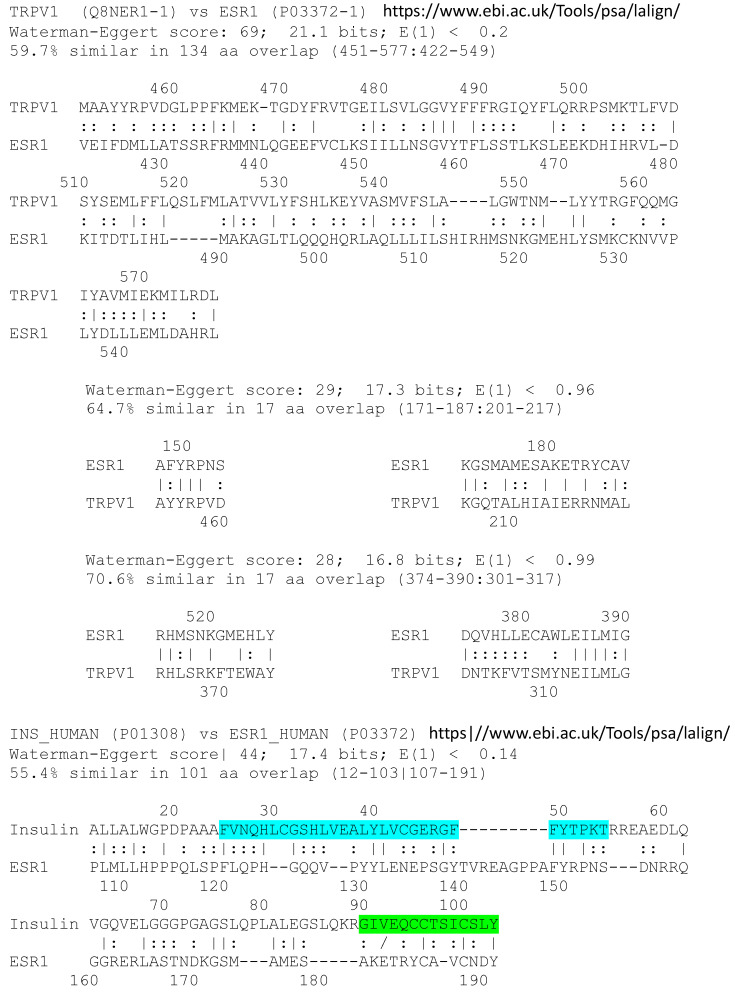
Paired similarities between the estrogen receptor (ESR1), TRPV1 and insulin, the insulin A chain (Ins A), or the insulin B chain (Ins B). Lines between the sequences indicate identical amino acids at those positions while dots between sequences indicate similar amino acids are present. The Ins A chain is highlighted in green within the entire insulin pre-pro-sequence at the top, while the Ins B chain is highlighted in blue.

**Figure 12 life-15-00208-f012:**
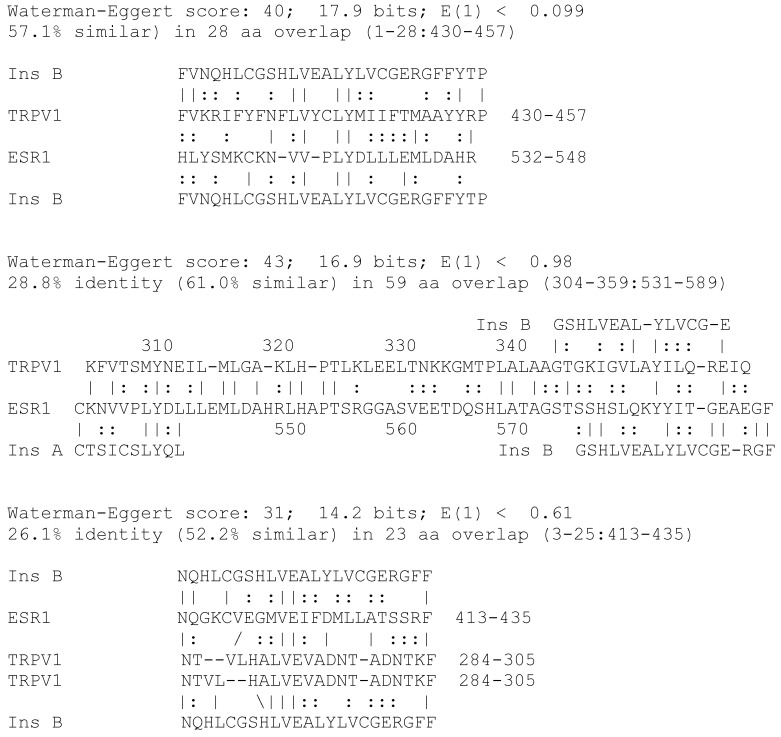
Common similarities shared between the estrogen receptor (ESR1), TRPV1 and the insulin A chain (Ins A), and the insulin B chain (Ins B). Lines between the sequences indicate identical amino acids at those positions while dots between sequences indicate similar amino acids are present.

**Figure 13 life-15-00208-f013:**
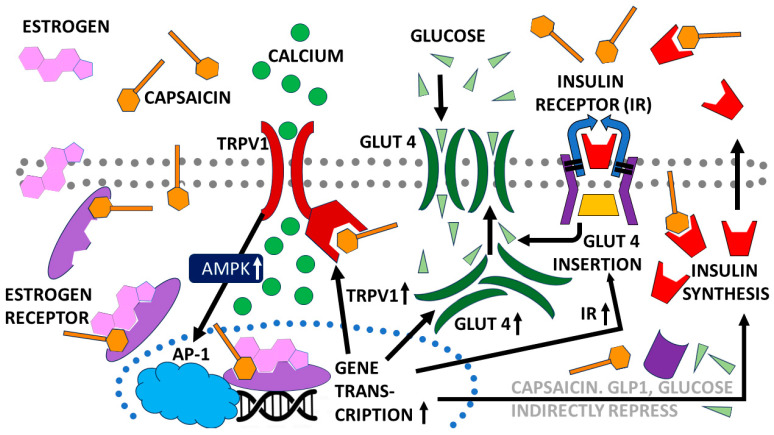
A schematic diagram integrating Figure 2 (main CAP energy-modifying mechanisms) with Figure 3 (main interactions between INS and estrogens) in light of the results reported here to illustrate the ways in which CAP, estrogen and insulins interact to modify energy production. Transient receptor potential vanilloid channel 1 (TRPV1); glucose transporter (GLUT); GLP1 (glucagon-like peptide 1); 5′-adenosine monophosphate protein kinase (AMPK); Activator Protein 1 (AP-1), a collective term referring to dimeric transcription factors composed of Jun, Fos or activating transcription factor. Large arrows indicate the direction of the response provoked by each interaction. Small upward-pointing arrows beside an acronym indicate that the activity of the molecule is increased.

**Table 1 life-15-00208-t001:** Summary of binding constants for capsaicin, vanillins and acetaminophen to estrogens and insulin.

Binding (Kd)	Capsaicin	Vanillin	Van Acid	Acetaminophen
Estradiol	No binding	No binding	No binding	No binding
Estriol	No binding	No binding	No binding	No binding
5-Pregnan-3β-OL-2O-one	No binding	No binding	No binding	No binding
Progesterone	No binding	No binding	No binding	No binding
4-Androstene-3,17-dione	No binding	No binding	No binding	No binding
Insulin	4 µM	50 µM	4 µM	70 µM

**Table 2 life-15-00208-t002:** Summary of known mechanisms by which CAP is thought to regulate energy consumption.

Article	Capsaicin Upregulates	Downregulates or Inhibits	Species
[26]	>Serum INS (CAP 6 mg/kg)	>Fasting blood glucose levels and serum level of glycosylated protein	T1DM rats
[28]	>Ca^2+^ ion release into the cell via TRPV-1 activation >Muscle actin–myosin interaction >Force generation >Intramuscular triglyceride consumption		Mice
[29]	>Fat metabolism through neuroadrenergic effects of TRPV-1 activation		Human (male)
[30]	>Sympathetic modulation (adrenal catecholamine secretion) and thermogenesis		Human (male)
[27]	>INS sensitivity—TRPV-1 activation stimulated GLP-1 release * >Fatty acid oxidation—modulated via TRPV1 activation >INS secretion from pancreatic β-cells ***	> Inflammatory factors **	Human * Mice ** Both ***
[31]	>Glucose uptake via AMPK activation and increasing p38 MAPK phosphorylation >Stimulation of ROS generation		Mice
[32]	>Activation of AMPK pathway	> Lipogenesis > Inhibition of AKT/mTOR pathway	Human HepG2 cells

**Table 3 life-15-00208-t003:** Summary of mechanisms previously established in the literature by which capsaicin may modulate cancer growth.

Article	Upregulates	Downregulates or Inhibits	Species	Cancer-Related Effects
[43]		>MDR-1 pathway → downregulates MDR-1 transcription (improves sensitivity)	Human HepG2 cells	>Sensitizes human cancer cells to cytocidal effects of chemotherapeutic agents >Selectivity to kill cancer cells with minimal damage to normal cells
[43]	>Bax/Bcl2 pathways → increased expression of Bax (apoptotic gene)	>Bax/Bcl2 pathway → decreased expression of Bcl-2 (prosurvival gene)	Human	>Induces apoptosis
[46]	>Increase in reduced glutathione and DPPH	>Cell cycle arrest in G0/G1 and G2/M phase >Suppresses ROS production	Human	>Inhibits cell proliferation
[44]	>Cleavage of procaspase-3 to caspase-3 >Interaction with caspase-1 >TRPV1-mediated Ca+ influx		Human (prostate cancer cells)	>Induces apoptosis
[51]	>Increases levels of LC3-II and Atg5 >Enhances p62 and Fap-1 degradation		Human nasopharyngeal carcinoma	>Regulation of autophagic pathways
[47]	>Activates p53-SMAR1 positive feedback loop >Decreases endothelial cell migration and network reformation	>VEGF expression	Human (NSCLC cells)	>Inhibition of tumor angiogenesis and metastasis
[52]		>Blocked metastatic burden >Mechanism unknown	Mice: TRPV1+ vs. TRPV1-	>Chemoprevention
[49]	>Increases estradiol, progesterone, CEA and malondialdehyde levels	>Decreased antioxidant enzymes	Rats (female)	>Blocks formation of nitrosomethylurea-induced mammary tumors
[53]	>Increases membrane fluidity		DPPC mem-brane model	
[48]			Mice	>Attenuates testosterone-induced prostate growth
[50]	>TGF-β1/Smad signaling pathway → expression of N-cadherin, vimentin, fibronectin >TRPV-1-mediated apoptosis	>TGF-β1/Smad signaling pathway → expression of E-cadherin, TGFBR2, TGF-β1 and p-Smad2/3 >IGF-1/AKT signaling pathway → expression of IGF-1, IGF-1R, p-AKT and RAGE >Androgen signaling pathway → expression of 5-*α*reductase type II, AR and PSA	Mice	>Ameliorated changes to the histological structure, prostate weight and prostate index (prostate/body weight ratio)

## Data Availability

All data are available by contacting the corresponding author (R.R.-B).

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
