# Peer review of "Capsaicin (But Not Other Vanillins) Enhances Estrogen Binding to Its Receptor: Implications for Power Sports and Cancers"

_life, 2025, doi:10.3390/life15020208_

Round 1
Reviewer 1 Report
Comments and Suggestions for Authors
Overall, the manuscript is satisfactory. Just few corrections should be taken care by author. They are listed below
References are not as per format, some where references with complete journal name where other places there is abbreviations.
Line 485: reference no. 24 Liang et al 2023, here journal name is abbreviated where in other places it is complete, kindly follow journal format, semicolon after 2023 where as in next ref in line 487 semincolon after jun 23. At some place month and date are written where they are not in other places. Kindly uniform the pattern of references as per journal
Line 138: ul is µl kindly correct it
Line 141, 149, 150, 165, 166, 168, 169, 170: ul is µl kindly correct it
Line 145: use full form of abbreviation where it is used first time EST-HRP. It is mentioned in Line: 216 estradiol linked to horse radish peroxidase
Line 135: ESR1 abbreviation and its full form at Line no 212. Kindly put it in Line 135 also
Line 200: why author using 205 and 210 nm absorbance for capsaicin, estrogen and estrogen receptor readings where as capsaicin possess 280 as lambda max and sometime at 210 nm, many molecules absorb strongly (e.g., nucleic acids, proteins, and other aromatic compounds). This overlap can make it challenging to distinguish between capsaicin, estradiol, and the estrogen receptor. Even many solvents, such as buffers and organic solvents also absorb strongly at this wavelength.
Line 66: explains CAP lowers total cholesterol whereas the ref. Abdalla 2022 says “…capsaicin plays a modulatory role in the biosynthesis and metabolism of a wider range of lipid signaling molecules in hTRPV1-HEK cells…” kindly check as the reference quote different cholesterol level in different cell types etc. kindly recheck.
Line 523: Self citated reference is not correctly written. In ref it is INS where as in publication it is Insulin. Kindly correct
Line 520: Ref. no 39 is missing in text and it is self citation pls see
Line 519: E coli must be written in italics.
Kindly reformat all the references as per journal format.
Comments on the Quality of English Language
English is ok
Author Response
Reviewer 1
Open Review
(x) I would not like to sign my review report
( ) I would like to sign my review report
Quality of English Language
( ) The quality of English does not limit my understanding of the research.
(x) The English could be improved to more clearly express the research.
|
Yes |
Can be improved |
Must be improved |
Not applicable |
|
|
Does the introduction provide sufficient background and include all relevant references? |
(x) |
( ) |
( ) |
( ) |
|
Is the research design appropriate? |
( ) |
(x) |
( ) |
( ) |
|
Are the methods adequately described? |
( ) |
(x) |
( ) |
( ) |
|
Are the results clearly presented? |
(x) |
( ) |
( ) |
( ) |
|
Are the conclusions supported by the results? |
(x) |
( ) |
( ) |
( ) |
Comments and Suggestions for Authors
Overall, the manuscript is satisfactory. Just few corrections should be taken care by author. They are listed below
References are not as per format, some where references with complete journal name where other places there is abbreviations.
Line 485: reference no. 24 Liang et al 2023, here journal name is abbreviated where in other places it is complete, kindly follow journal format, semicolon after 2023 where as in next ref in line 487 semincolon after jun 23. At some place month and date are written where they are not in other places. Kindly uniform the pattern of references as per journal
The journal editors will make uniform after acceptance. This is not a problem.
Line 138: ul is µl kindly correct it: Line 141, 149, 150, 165, 166, 168, 169, 170: ul is µl kindly correct it
All corrected.
Line 145: use full form of abbreviation where it is used first time EST-HRP. It is mentioned in Line: 216 estradiol linked to horse radish peroxidase
Corrected
Line 135: ESR1 abbreviation and its full form at Line no 212. Kindly put it in Line 135 also
Corrected.
Line 200: why author using 205 and 210 nm absorbance for capsaicin, estrogen and estrogen receptor readings where as capsaicin possess 280 as lambda max and sometime at 210 nm, many molecules absorb strongly (e.g., nucleic acids, proteins, and other aromatic compounds). This overlap can make it challenging to distinguish between capsaicin, estradiol, and the estrogen receptor. Even many solvents, such as buffers and organic solvents also absorb strongly at this wavelength.
This is now addressed in the Methods section. The fact that many solvents and buffers absorb at 205-210 nm is irrelevant to the methodology used. To begin with, only one solvent was used for all of the studies, so that did not vary. Secondly, the solvent absorbance was measured by itself and subtracted from the solvent-compound mixtures so that the calculations of variance in absorbance from expected values were all made from the compound-only absorbances. Third, pure chemicals were used so that there were no nucleic acids, proteins, or other aromatic compounds in our solutions that could interfere with our results.
Line 66: explains CAP lowers total cholesterol whereas the ref. Abdalla 2022 says “…capsaicin plays a modulatory role in the biosynthesis and metabolism of a wider range of lipid signaling molecules in hTRPV1-HEK cells…” kindly check as the reference quote different cholesterol level in different cell types etc. kindly recheck.
The Reviewer has taken one line out of a very long and complex review. Yes, it is correct that capsaicin works through different mechanisms in different types of cells but if the Reviewer uses the word “capsaicin” to search through the review, s/he will find that in every case, in every type of cell, capsaicin lowers accumulation of fats, including cholesterol. Thus, our statement is correct: “cumulative experimental evidence showed that expression and/or activity of TRPV1 is increased in metabolic syndromes, such as in artherosclerosis, obesity, hyperlipidemia and NAFLD, indicating that TRPV1 is involved in the signaling pathway that favors fat accumulation. Thus, shutting down of this signaling pathway with TRPV1 antagonists or agonists that cause desensitization may counteract fat accumulation.”
Line 523: Self citated reference is not correctly written. In ref it is INS where as in publication it is Insulin. Kindly correct
Corrected.
Line 520: Ref. no 39 is missing in text and it is self citation pls see
This is not so: it is cited on line 237 (try searching on “Churchill”)
Line 519: E coli must be written in italics.
Corrected.
Kindly reformat all the references as per journal format.
As noted above, this is not something that we, as authors need to do: the journal editors will take care of this during the publication processing.
Comments on the Quality of English Language
English is ok
Submission Date
19 December 2024
Date of this review
12 Jan 2025 17:44:16
Bottom of Form
© 1996-2025 MDPI (Basel, Switzerland) unless otherwise stated
Disclaim
Reviewer 2 Report
Comments and Suggestions for Authors
The article is interesting and essential in the cognitive context.
However, I wonder whether the extensive Introduction with many interesting figures does not distort the proportion.
This is neither a review nor an article—something in between.
The figures from the Introduction could be moved to the Appendix or a different place because the article is overwhelming in the current version.
Figures 11-13 from the discussion should be moved to the supplement. The tables should not be in the discussion section.
The article is intriguing and valuable, but the authors must work on the message. The number of figures should be reduced. Maybe some should be moved to the supplement or, better yet, to the Appendix.
The conclusions are missing. They need to be supplemented.
The article should be organized.
The article's content is valuable, and the message is essential, but it needs to be more coherent and organized.
Author Response
Reviewer 2
Open Review
(x) I would not like to sign my review report
( ) I would like to sign my review report
Quality of English Language
(x) The quality of English does not limit my understanding of the research.
( ) The English could be improved to more clearly express the research.
|
Yes |
Can be improved |
Must be improved |
Not applicable |
|
|
Does the introduction provide sufficient background and include all relevant references? |
( ) |
( ) |
(x) |
( ) |
|
Is the research design appropriate? |
( ) |
(x) |
( ) |
( ) |
|
Are the methods adequately described? |
( ) |
(x) |
( ) |
( ) |
|
Are the results clearly presented? |
( ) |
(x) |
( ) |
( ) |
|
Are the conclusions supported by the results? |
( ) |
( ) |
(x) |
( ) |
Comments and Suggestions for Authors
The article is interesting and essential in the cognitive context.
However, I wonder whether the extensive Introduction with many interesting figures does not distort the proportion.
This is neither a review nor an article—something in between.
The author is correct: In order to understand our hypothesis, it is necessary to review several different lines of research and then to show how they intersect in a way that leads to our novel hypothesis. There is no way to motivate the hypothesis except to summarize each of the currently independent lines of research and how they unexpectedly intersect. This is an unavoidable consequence of doing integrative research.
The figures from the Introduction could be moved to the Appendix or a different place because the article is overwhelming in the current version.
Two other reviewers found the Introduction to be very good. Besides which, the figures will not decrease the amount of information in the Introduction – they are actually brief summaries of the text! We prefer to leave as is, especially since some people prefer visual forms of information to verbal or written ones.
Figures 11-13 from the discussion should be moved to the supplement. The tables should not be in the discussion section.
They are essential to validating the idea that capsaicin can bind to very different proteins because these proteins share small, highly similar modules. We have therefore moved them instead (in accordance with Reviewer 4’s suggestion) to the Results.
The article is intriguing and valuable, but the authors must work on the message. The number of figures should be reduced. Maybe some should be moved to the supplement or, better yet, to the Appendix.
The conclusions are missing. They need to be supplemented.
Conclusions are not required but given this Reviewer’s concern and that of Reviewer 3 that the overall message is not clear, we have tried to provide that clarity in a new conclusion section.
The article should be organized.
Again, other reviewers found it well organized and we choose not to re-organize it.
The article's content is valuable, and the message is essential, but it needs to be more coherent and organized.
Submission Date
19 December 2024
Date of this review
30 Dec 2024 22:26:23
Reviewer 3 Report
Comments and Suggestions for Authors
Dears authors,
The manuscript “Capsaicin (but Not Other Vanillins) Enhances Estrogen Binding to Its Receptor: Implications for Power Sports and Cancers” provides a comprehensive understanding of the critical role of different bioactive compounds in mechanisms related to sports and cancer. Though this article is very informative, it has some limitations that the authors describe well. It would be interesting to see if the effects remained the same in vivo studies.
So, I have some observations regarding the article form. I suggest placing the first proposition of the introduction section at the end.
At rest, the introduction is well-organized and informative.
However, from my point of view, the methods section can be improved. Also, you can add the software that you used for the diagrams.
Some figures, like 11 and 12, could be improved by being arranged more efficiently in the article text.
The Discussion section is very well presented, but I didn't find the Conclusions section.
Considering the results, you could present some future perspectives based on them.
Author Response
Reviewer 3
Open Review
( ) I would not like to sign my review report
(x) I would like to sign my review report
Quality of English Language
(x) The quality of English does not limit my understanding of the research.
( ) The English could be improved to more clearly express the research.
|
Yes |
Can be improved |
Must be improved |
Not applicable |
|
|
Does the introduction provide sufficient background and include all relevant references? |
(x) |
( ) |
( ) |
( ) |
|
Is the research design appropriate? |
(x) |
( ) |
( ) |
( ) |
|
Are the methods adequately described? |
( ) |
(x) |
( ) |
( ) |
|
Are the results clearly presented? |
(x) |
( ) |
( ) |
( ) |
|
Are the conclusions supported by the results? |
( ) |
(x) |
( ) |
( ) |
Comments and Suggestions for Authors
Dears authors,
The manuscript “Capsaicin (but Not Other Vanillins) Enhances Estrogen Binding to Its Receptor: Implications for Power Sports and Cancers” provides a comprehensive understanding of the critical role of different bioactive compounds in mechanisms related to sports and cancer. Though this article is very informative, it has some limitations that the authors describe well. It would be interesting to see if the effects remained the same in vivo studies.
So, I have some observations regarding the article form. I suggest placing the first proposition of the introduction section at the end.
Done.
At rest, the introduction is well-organized and informative.
However, from my point of view, the methods section can be improved. Also, you can add the software that you used for the diagrams.
We have expanded the Method section to be more precise and complete in our description of the experiments and added the software used for the diagrams.
Some figures, like 11 and 12, could be improved by being arranged more efficiently in the article text.
We have moved these to the end of the Results section.
The Discussion section is very well presented, but I didn't find the Conclusions section. Considering the results, you could present some future perspectives based on them.
As noted for Reviewer 2, Conclusions are not required but given both Reviewer’s concern that the overall message is not clear, we have tried to provide that clarity in a new conclusion section.
Submission Date
19 December 2024
Date of this review
06 Jan 2025 11:40:44
Reviewer 4 Report
Comments and Suggestions for Authors
Comments and Suggestions for Authors
The article entitled; “Capsaicin (but Not Other Vanillins) Enhances Estrogen Binding to Its Receptor: Implications for Power Sports and Cancers” is interesting and scientifically sound. However, I think, this research has rough results and is not in depth. The author should add more information or do more works to make it more interesting and suitable for publication. Only some parts of your results that aligned with your purpose. It is recommended to add more work, such as investigating the mechanisms of how the substances interact with different receptors (mode of action). Additionally, I would suggest that the authors should clarify some mistakes in the remarks below:
Remarks:
- Page 1/23, Line 26-27
Keywords: capsaicin; acetaminophen; estradiol; estrogen receptor; insulin; glucose transporter
energy production ### I think it should be revised, some words are unsuitable to be a keyword, the other words are more suitable i.e. acetaminophen, energy production should be substituted to anti-cancer.
- Page 1/23, Line 30-32
The purpose of this study is to investigate some novel sex-hormone-related mechanisms of capsaicin and other vanilloid compounds that may influence their acute effects on glucose metabolism.
### From the result, I did not find the data that supported this purpose.
- Page 2/23, Line 49
…… (reviewed in [Liang, et al., 2023]) ……
### reviewed in, should be deleted, only citation is enough. Please correct throughout your article.
- Page 5/23,
2.MATERIALSANDMETHODS
### I think, the authors should provide more information about the source of chemical compounds as you used in this experiment i.e. capsaicin, estradiol, all vanilloid compounds and so on.
- Page 7/23, Line 203, 206
…… CAP did bind to the ESR1…..with lower affinity than did CAP..
### What does it mean?
- Page 7/23, Figure 7.
.01 mg/m. CAP ### Please, correct it to .01 mg/ml
- Page 9/23, Figure 8.
Blnding ### Binding
- Figure 8 Effect of Vanillin on Estrogen-HRP Binding to the Estrogen Receptor
and Figure 9 Effect of Acetaminophen on Estradiol Binding to the Estrogen Receptor
### How different of Estrogen-HRP and Estradiol Binding to the Estrogen Receptor?
- Figure 10 Comparison of the effects of different vanillin-like compounds on the binding of estradiol conjugated to horse radish peroxidase (EST-HRP) to the estrogen receptor.
### Why is there no comparison with the effects of capsaicin?
- Table 1. Summary of binding constants for capsaicin, vanillins and acetaminophen to estrogens and insulin.
### I think Table 1 should be rewritten to make it easier to understand.
- Page 11/23, Line 251-258
### It should be written in a way that is easier to understand. Write it in complete sentences, not as a short message.
Cheers,
Date of this review
11 January 2025
Author Response
Reviewer 4
Open Review
(x) I would not like to sign my review report
( ) I would like to sign my review report
Quality of English Language
(x) The quality of English does not limit my understanding of the research.
( ) The English could be improved to more clearly express the research.
|
Yes |
Can be improved |
Must be improved |
Not applicable |
|
|
Does the introduction provide sufficient background and include all relevant references? |
( ) |
(x) |
( ) |
( ) |
|
Is the research design appropriate? |
( ) |
( ) |
(x) |
( ) |
|
Are the methods adequately described? |
( ) |
( ) |
(x) |
( ) |
|
Are the results clearly presented? |
( ) |
( ) |
(x) |
( ) |
|
Are the conclusions supported by the results? |
( ) |
( ) |
(x) |
( ) |
Comments and Suggestions for Authors
Comments and Suggestions for Authors
The article entitled; “Capsaicin (but Not Other Vanillins) Enhances Estrogen Binding to Its Receptor: Implications for Power Sports and Cancers” is interesting and scientifically sound. However, I think, this research has rough results and is not in depth. The author should add more information or do more works to make it more interesting and suitable for publication. Only some parts of your results that aligned with your purpose. It is recommended to add more work, such as investigating the mechanisms of how the substances interact with different receptors (mode of action).
The suggestion of looking into interactions with “different receptors” to determine “mode of action” is rather vague. What other receptors and why those in particular? More to the point, we proposed a very specific hypothesis at the end of the Introduction which has guided the specific experiments that we did perform. It is not clear from our hypothesis that other receptors would be relevant.
Additionally, I would suggest that the authors should clarify some mistakes in the remarks below:
Remarks:
- Page 1/23, Line 26-27
Keywords: capsaicin; acetaminophen; estradiol; estrogen receptor; insulin; glucose transporter
energy production ### I think it should be revised, some words are unsuitable to be a keyword, the other words are more suitable i.e. acetaminophen, energy production should be substituted to anti-cancer.
Anti-cancer substituted for energy production.
- Page 1/23, Line 30-32
The purpose of this study is to investigate some novel sex-hormone-related mechanisms of capsaicin and other vanilloid compounds that may influence their acute effects on glucose metabolism.
### From the result, I did not find the data that supported this purpose.
Well, maybe that is because the Reviewer has attributed to us a goal which is not what we set out to do. At the end of the Introduction, we state, we believe clearly, that: “we hypothesize that the observed sex-dependent differences in CAP effects may be mediated by direct interactions with the estrogen and INS-glucose systems” and we’re pretty certain that is what this paper does. We’ve clarified this further by modifying the hypothesis to read: …
that capsaicin’s sex-related metabolic effects may be due to a direct interaction with insulin, estrogens, and/or their receptors.
- Page 2/23, Line 49
…… (reviewed in [Liang, et al., 2023]) ……
### reviewed in, should be deleted, only citation is enough. Please correct throughout your article.
Corrected.
- Page 5/23,
2.MATERIALSANDMETHODS
### I think, the authors should provide more information about the source of chemical compounds as you used in this experiment i.e. capsaicin, estradiol, all vanilloid compounds and so on.
Added.
- Page 7/23, Line 203, 206
…… CAP did bind to the ESR1…..with lower affinity than did CAP..
### What does it mean?
As the Reviewer is surely aware, the Results section is not the place where the meaning of experiments is discussed. The meaning of the lower affinity of vanilloid compounds other than CAP binding to the ESR is discussed at length in the Discussion and is done so, more properly, in light of the other types of experiments that were performed, which is, in this case, very important since the binding of the vanilloid compounds to the ESR has different effects on estradiol binding depending on the compound. The differences in binding affinities to the ESR1 by the various compounds does not directly predict whether they will enhance, inhibit or have no effect on estradiol binding. We note this in the Conclusion as well.
- Page 7/23, Figure 7.
.01 mg/m. CAP ### Please, correct it to .01 mg/ml
Corrected.
- Page 9/23, Figure 8.
Blnding ### Binding
Corrected
- Figure 8 Effect of Vanillin on Estrogen-HRP Binding to the Estrogen Receptor
and Figure 9 Effect of Acetaminophen on Estradiol Binding to the Estrogen Receptor
### How different of Estrogen-HRP and Estradiol Binding to the Estrogen Receptor?
The “control” in the figure is estrogen-HRP binding to the estrogen receptor in the absence of vanillin or acetaminophen. This has now been clarified in the figure captions.
- Figure 10 Comparison of the effects of different vanillin-like compounds on the binding of estradiol conjugated to horse radish peroxidase (EST-HRP) to the estrogen receptor.
### Why is there no comparison with the effects of capsaicin?
The figure was designed to compare the decrease in binding caused by vanillin-like compounds but the Reviewer has a point that there is no reason to leave out capsaicin, which has now been added for comparison.
- Table 1. Summary of binding constants for capsaicin, vanillins and acetaminophen to estrogens and insulin.
### I think Table 1 should be rewritten to make it easier to understand.
The “>1 mM” entries have been replaced with “No binding”, with an explanation in the caption that the latter indicates the former.
- Page 11/23, Line 251-258
### It should be written in a way that is easier to understand. Write it in complete sentences, not as a short message.
Done!
Cheers,
Date of this review
11 January 2025
Submission Date
19 December 2024
Date of this review
11 Jan 2025 14:50:43
Round 2
Reviewer 2 Report
Comments and Suggestions for Authors
After reviewing the authors' responses and the changes made to the article, I was convinced by the authors that the layout they proposed was reasonable.
I therefore believe that the article in its current form does not require any further changes and can be processed in its current form.
Reviewer 4 Report
Comments and Suggestions for Authors
The author has made an effort to clarify and add more information in the article to enhance its clarity and understanding. Therefore, it is appropriate to publish this work in the journal Life.